# Medication-Related Complaints in Residential Aged Care

**DOI:** 10.3390/pharmacy11020063

**Published:** 2023-03-23

**Authors:** Juanita L. Breen, Kathleen V. Williams, Melanie J. Wroth

**Affiliations:** 1Clinical Pharmacy Unit, Aged Care Quality and Safety Commission, Canberra, ACT 2601, Australia; 2Wicking Dementia Centre, College of Health and Medicine, University of Tasmania, Hobart, TAS 7001, Australia; 3Chief Clinical Advisor, Aged Care Quality and Safety Commission, Parramatta, NSW 2124, Australia

**Keywords:** medication, medicine, complaints, long-term aged care, nursing home, opioids, psychotropic, prescribing regulations, medication safety, chemical restraint

## Abstract

Complaints reflect a person’s or family’s experience within the aged care system and provide important insight into community expectations and consumer priorities. Crucially, when aggregated, complaints data can serve to indicate problematic trends in care provision. Our objective was to characterize the areas of medication management most frequently complained about in Australian residential aged care services from 1 July 2019 to 30 June 2020. A total of 1134 complaint issues specifically referenced medication use. Using content analysis, with a dedicated coding framework, we found that 45% of these complaints related to medicine administration processes. Three categories received nearly two thirds of all complaints: (1) not receiving medication at the right time; (2) inadequate medication management systems; and (3) chemical restraint. Half of the complaints described an indication for use. These were, in order of frequency: ‘pain management’, ‘sedation’, and ‘infectious disease/infection control’. Only 13% of medication-related complaints referred to a specific pharmacological agent. Opioids were the most common medication class referred to in the complaint dataset, followed by psychotropics and insulin. When compared to complaint data composition overall, a higher proportion of anonymous complaints were made about medication use. Residents were significantly less likely to lodge complaints about medication management, probably due to limited engagement in this part of clinical care provision.

## 1. Introduction

‘Medication’ is any drug or preparation that is used to prevent, treat, and cure illness [1], and includes prescription and non-prescription medicines, and complementary health care products [2] (p. 7). ‘Medication administration’ refers to the actual giving of medicine [3] and ‘medication management’ involves the processes of prescribing, dispensing, supplying, administering, and monitoring medicines [4].

Increasingly, older Australians are supported to live in their own homes for as long as possible. This means that those who do move into residential aged care services (RACSs) are older and frailer, and will often have several chronic health conditions requiring a high level of care and multiple medications [5]. A recent study found that over 75% of Australian aged care residents were prescribed 10 or more medications [6]. The more medications a person is prescribed, the higher the risk of adverse events, drug interactions, and other poor health outcomes.

Over three quarters of aged care residents require help to take medicines [7]. Staff administering medication are required to do so in accordance with national guidelines, State or Territory law and the Aged Care Quality Standards [5,8]. Medication administration and management involves multiple components and various health practitioners. At an organizational level, the systems and practices of prescribing, ordering, preparing and dispensing, supplying, storing, administering, and monitoring medicines in RACSs involves a range of prescribers, pharmacists, nursing and care staff, residents, relatives, and/or residents’ substitute decision-maker [9]. As with any complex process, problems can occur at any stage [5].

Issues with medication administration and management in RACSs can lead to severe consequences, including hospital admission and death. A Western Australian study found that 20% of unplanned hospital admissions for aged care residents were a direct result of inappropriate medication use [10]. Furthermore, there have been at least 30 coronial investigations into medicine-related deaths occurring in aged care since 2000, with the majority attributed to medication administration and monitoring errors [11].

Anybody with a concern or complaint relating to Commonwealth-subsidized aged care, that they have not been able to resolve by talking with the service-provider, can contact the Aged Care Quality and Safety Commission (the Commission) for advice and assistance or to lodge a complaint [12]. All complaints received by the Commission are registered and recorded on a standardized form in a relational database, the ‘National Complaints and Compliance Information Management System’ (NCCIMS) [12]. Given the number of consumers administered multiple medications and the complexity of medication management processes, it is not surprising that complaints relating to medication occur frequently. In the 2017–2018 and the 2018–2019 Commission annual reports, medication-related complaints were the most frequent complaint listed, comprising over 7% of all complaint issues [13,14]. It is important to understand that each complaint can include more than one issue. Throughout 2019–20 over 15,600 separate complaint issues in Australian RACSs were recorded in 6335 complaints [15].

Residents and their families and/or representatives, staff, and other observers are sensitive to, and able to recognize, a range of problems in healthcare delivery, some of which are not identified by other monitoring systems, such as incident reporting systems, case reviews, or academic research. Crucially, when complaint data are aggregated they can indicate trends in healthcare provision [16]. In addition, residents, their family, and representatives will identify and prioritize issues most relevant to them; thus, complaint data can provide additional information for RACSs on how to improve the aspects of care most important to their consumers, including communication and patient safety. Furthermore, analyzing data on poor consumer experiences provides a strong incentive for organizations to identify systematic problems in care, learn from them, reflect on the contributing components, and put measures in place to improve care provision and prevent possible adverse events in the future [16,17,18].

The aim of this evaluation project was to identify, categorize, and quantify the main complaint issues lodged in relation to medication management and administration in Australian residential aged care during 2019–2020. A secondary aim was to determine the participant groups or complainant type making medication-related complaints. Based on the findings, key recommendations are made to improve the quality of medication use in RACSs.

## 2. Materials and Methods

### 2.1. Complaints Dataset

Anyone with a concern about residential aged care can contact the Commission by telephone, in writing, or online. Where a complainant makes contact by telephone, Commission officers support the person with information and options to resolve the issue informally. When first contacted, Commission officers ask questions to understand the issue/s and expectations of the person with the concern, and advise if they can assist [12].

The person may decide to lodge a formal complaint with the Commission. Complainants are asked to provide concise information about the issue, describe persons involved, and submit any documentation relating to the complaint (Appendix A lists the questions posed). Upon receipt, trained complaints officers assess what issues are raised, record this information in the NCCIMS database, and discuss options and the complaints process with the complainant. Each complaint issue is itemized, recorded, and then allocated a keyword and sub-keyword as descriptors in the NCCIMs database. No personal information is disclosed unless authorized by the complainant and all data are stored in accordance with the *Privacy Act 1998* and Australian privacy principles [12].

The complainant has the option of complaining openly, confidentially, or anonymously. When the complaint is submitted openly, the complainant’s name and contact details are provided to the Commission, which will pass them on to the service-provider when the Commission contacts them to discuss the complaint. If the complaint is lodged confidentially, the Commission will not disclose the identity of the complainant to the service-provider but can contact the complainant for further information if needed and keep them informed about the complaint progress or outcome. If a complaint is lodged anonymously, the Commission and provider do not know the identity of the complainant; the complainant cannot be informed of the complaint’s progress or outcome, and is unable to provide further data [12].

To ensure all medication-related complaints were identified, the full list of NCCIMs keywords and associated sub-keywords was examined by a pharmacist researcher (JB), senior geriatrician (MW), and health project officer (KW), who discussed and then nominated the NCCIMS descriptors most likely to involve medication-related issues (Appendix A). The decision was made to not include the sub-keyword complaint issues of pressure care, wound management, and skin care because it was considered that medication would not feature strongly in these areas. A Commission information analyst then prepared an anonymized dataset of the selected keywords and sub-keyword complaint issues from 1 July 2019 to 30 June 2020. All identifying information was removed. The dataset for this evaluation included complainant type which are representative or family member, anonymous, care recipient, other interested (staff from the service), external agency (other organizations such a hospital, advocacy service, and guardianship board) and internal agency (from another section of the Commission); and complaint issue description of what exactly occurred, when it occurred, persons involved, and other relevant anonymized case details, such as consequences resulting from the concern [12].

To begin, duplicate issues were removed from the dataset. These included complaints allocated the same registration number with several entries made, or complaints that replicated content about the same issue that may have been lodged online by the same complainant on different dates. Each issue description and accompanying case detail field was then examined to assess if it was directly related to medication administration and management. Issues related to oxygen administration and devices used to manage sleep apnea were excluded at this stage, as were complaints not specifically referring to the use of medication. For example, general complaints referring to poor management of behavior or dementia were not included unless medication treatment was specifically mentioned.

The number of complaint issues were then tallied to determine the keywords and sub-keywords utilized by complaint officers to describe complaints involving medication.

### 2.2. Qualitative Analysis

Each complaint issue was described and recorded by the complaints officer using the NCCIMS database. Two of the NCCIMS data-fields provided descriptive detail about the nature of the complaint issue, any persons involved, and consequences resulting from the issue, where relevant. To gain in-depth understanding around the medication-related complaints, these written data-fields were examined employing qualitative analysis that involved both inductive and deductive approaches. Content analysis was used to identify and interpret the medication-related complaints dataset by categorizing small pieces of the data that represent relevant concepts [17]. To ensure a systematic and detailed analysis, a dedicated coding framework was created based on relevant NCCIMS keywords and sub-keywords, and external medication administration and management classifications. The framework coding categorization (coding framework) was developed, trialed, and then reviewed by JB and KW.

The coding framework used to categorize the medication complaint dataset draws on relevant NCCIMs keywords and sub-keywords but also incorporates the six rights of medication administration as described in the ‘National Residential Medication Chart’ which was developed to ensure consistent and rigorous reporting of medicines management and administration in Australian residential aged care (Appendix A) [19]. The ‘six rights of medication administration’ ensure that the ‘right resident’ receives the ‘right medicine’, at the ‘right dose’, by the ‘right route’, and at the ‘right time’. Having ‘the right documentation’ is the important sixth right [19].

The six rights form the first six categories, and categorize complaint issues relating to the keyword of ‘Health Care’ and the sub-keyword of ‘medication administration and management’. A total of 13 categories were set as the coding framework. These categories were broken down further into 26 ‘complaint issue’ codes (the issue codes can be seen in Table 1 in the Results section). All complaint issues were individually examined by JB and coded into one of these 26 separate issue codes to precisely describe the different aspects of medication administration and management to which each complaint was related. Once the dataset was fully coded, all codes were aggregated. The frequency of different complaint issues was tallied and recorded using basic manual counts, and the predominant categories determined. Most complaint issues could be allocated to a single code; however, 90 complaint issues raised more than one medication-related concern so were duplicated and then coded to two or more of the appropriate issue codes. Examples of complaint issues were selected and used to demonstrate the medication areas receiving the most complaints. These issues are italicized and quoted as described in the NCCIMs database in the Results section.

The complaints dataset was also examined to ascertain if there were certain medical conditions, medication classes, or individual medicines about which complaints had been lodged. In the Australian acute health sector, the ‘APINCHS’ acronym and classification (Figure 1) assists clinicians to focus on a group of medicines known to be associated with high potential for medication-related harm [20,21]. As a final step, the APINCHS categorization system was used to assess the proportion of high-risk medication use referred to in the medication-related complaints dataset [21].

### 2.3. Statistical Analysis

To assess if the type of participants or cohort making medication-related complaints differed from the general distribution of all complainant types lodging complaints about residential aged care, the distribution was compared to overall complainant data obtained from the 2019–2020 Commission Annual Report [15]. These associations were analyzed using Chi-square tests via the MedCalc© (Version 20.218) comparisons of proportions calculator [22].

## 3. Results

A total of 7859 complaint issues were extracted to form the original medication-related dataset. First, duplicates were removed, then 4023 complaint issues were individually examined to remove issues considered not relevant to medication. Figure 2 illustrates how the final medication-related dataset of 1134 complaint issues evolved.

### 3.1. Keywords/Sub-Keywords Used to Describe Medication-Related Complaints

Table 1 below sets out the NCCIMS keywords and sub-keywords used by complaint officers to describe issues raised about medications in residential aged care from 1 July 2019 to 30 June 2020. The three sub-keywords: ‘physical restraint’, ‘mental health’, and ‘sleep’ listed in Appendix A were not included as there were no medication-related complaint issues reflecting these fields.

Although most of the medication-related complaint issues *(n* = 824, or 73%) were described in NCCIMS using the sub-keyword ‘medication administration and management’, more than a quarter of medication-related issues (27%) were coded by complaint officers under alternative NCCIMS sub-keywords categories, such as ‘chemical restraint’, ‘pain management’, ‘information about medication’, and ‘infectious disease’.

### 3.2. Types of Medication-Related Complaint Issues

Using the coding framework, all 1134 complaint issues were individually examined and coded to one of 26 separate issue codes. The descriptions of the medication-related complaint issues varied markedly from a single concise sentence of 10 words to detailed descriptions exceeding 400 words in length. Ninety of the medication-related complaint issues contained two or more codes; for example, one complaint issue stated:


*‘The service has inadequate medication management procedures as staff are unfamiliar with S4 and S8 medications, drug charts are difficult to read, observations are not completed, and some medications (opioids) are not administered overnight because there is no RN’.*
(registered nurse)

This single complaint issue was coded to the issue codes of ‘right time’ (medications given late) and ‘appropriate systems, processes, and policies in place’ (provider processes and policies). In several of the complaints, three or more issues were identified in the one complaint, thus 1224 separate complaint issues were coded in total. The coding framework, types, and number of issues are presented in Table 2, with codes tallying more than 50 complaints emboldened for emphasis.

Nearly half of all complaint issues related to the six rights of administration (n = 545 or 45%) [19]. Three of the 13 categories (two sub-keywords had the same ‘appropriate policies/systems’ category) covered the majority (n = 751 or 61%) of all complaint issues; specifically:Not receiving medication at the right timeInadequate medication management systems, processes or policies in placeChemical restraint.

#### 3.2.1. Not Receiving Medication on Time

The most frequent issue, accounting for over a quarter of complaints (n = 312 or 25%) was not receiving medicines on time. These types of complaints involved doses missed or given too late, or that treatment was significantly delayed, or not started at all:


*‘Concern during a scheduled 4-week respite stay the service failed to provide appropriate medication management to consumer as his medication provided to the service upon admission was left untouched.’*


Many of these complaints, especially those related to analgesia, led to significant distress:


*‘Service did not provide adequate pain management. Specifically, there was a delay of 4 hours before morphine was given to her after the doctor had prescribed it at 1 pm. As a result, the resident was screaming in pain.’*


#### 3.2.2. Inadequate Medication Management Systems, Policies and Procedures

The second most frequent complaint category related to a lack of ‘appropriate systems, policies, and procedures’, accounting for over a fifth of all of all medication-related complaints (n = 271 or 22%). Most of the complaint issues reported were non-specific or generally worded complaints about the provider’s inadequate medication management processes and procedures, for example:

*‘Medication management at the service simply does not meet the basic needs of its consumers’,* and
*‘Concern about the appropriate identification, handling, timing and management of medication.’*

Inadequate qualifications of staff administering medication was a significant issue of concern in this category, accounting for 65 complaints or 5% of complaints overall:

*‘Registered nurses are not checking medications with another registered nurse but instead relying on PCA (personal care assistants) to check dosages.’* and *‘Kitchen staff are administering medications and they are not trained to do so.’*

Several complaints refer to unlawful practice against drug and poisons regulation, often involving unqualified and untrained staff:


*‘Enrolled nurses are administering Schedule 8 drugs to consumers unsupervised and on a daily basis.’*
(Schedule 8 drugs are controlled drugs of dependence, mainly opioids. Legally they must only be administered and documented by a registered nurse)

#### 3.2.3. Chemical Restraint

The third most common complaint issue was the use of medication to limit movement or control behavior, a restrictive practice defined as ‘chemical restraint’ [23]. This issue code received 168 complaints, accounting for 14% of all medication-related complaint issues. Most of the issues described include complaints around the general overuse of these medications:

*‘The residents are being inappropriately medicated with either painkillers or psychotropics.’* and *‘It is a regular occurrence that you see a resident wandering around or being disruptive, something might happen and then the next time you see them they are heavily sedated.’*

Some of the complaints involved residents suspecting they were given medication to sleep or calm them down without obtaining their consent:


*‘Consumer is unaware of what medication she is given, especially at night, and she often wakes up groggy, and has expressed concern that she was given sleeping tablets to prevent her from getting up during the night to use the toilet.’*


In contrast, several of the complaints refer to the consequences of reducing these types of medications without using alternative management strategies:


*‘Care recipients have ceased medications that may be classified as a form of restraint, which has led to an increase in unmanaged care recipient aggression.’*


#### 3.2.4. Other Complaint Issues

Two other categories of medication-related issues received over 50 complaints. First, those involving a resident’s right to refuse medication, specifically the ‘lack of informed consent’ before medicines were started, altered, or ceased (n = 65). Notably, over a third of these complaints (n = 24) relate to medication that may be used as chemical restraint:


*‘The consumer is being sedated against the wishes of her Enduring Power of Attorney. She is being administered risperidone (antipsychotic) with no diagnosis, medical reason and no explanation to her family.’*


The other complaint issue that featured prominently related to staff administering incorrect doses of a medicine (n = 53):


*‘A concern that resident was being given inconsistent and incorrect doses of his medications while at the nursing home, resulting in repeated, unnecessary hospital admissions.’*


Medicines were either under-dosed, over-dosed, or administered simultaneously when one medication was meant to be ceased before another was started:


*‘Instead of ceasing medication for depression and replacing it with another, the resident was given both medications for 3 months.’*


### 3.3. Medical Conditions and Medicines Associated with Medication-Related Complaint Issues

Of the 1134 medication-related complaint issues, 554 (49%) mention the medical condition or indication for the medicine (Table 3). The medical conditions or indications about which the highest number of complaints were received involved ‘pain management and/or palliative care’.

Many of these complaints were about delayed timing of treatment and/or a lack of qualified staff to administer treatment:


*’The family were asked would they be happy to let the consumer go 12 hours without a morphine injection (until the RN came back on duty in the morning).’*


The use of medications for the intention of sedation was also a frequent complaint, accounting for nearly a third of all medical conditions/indications that were specifically identified. Infectious disease management, especially a delay to, or absence of, treatment of urinary tract infections (UTIs) were also a frequent concern, accounting for over 50 separate complaint issues. A significant proportion of these complaints were about urine testing for UTIs, which may indicate a lack of awareness of, and communication around, guidelines for UTI testing in older people in the context of antimicrobial stewardship:


*‘Staff have not taken action to obtain a urine specimen from mum despite her daughter requesting a simple ‘dip stick’ be done to check if there is an indication of a UTI.’*


Diabetes and Parkinson’s disease were two other medical conditions frequently highlighted in the complaints database. Complainants were concerned about the administration of insulin injections and about poor blood glucose control in general. The overwhelming number of complaints cited for Parkinson’s disease related to late administration of dopaminergic medications, or doses being missed altogether, which can lead to residents ‘freezing up’, or being unable to move:


*‘The service does not provide my mum with her Parkinson’s medication before her shower. As a result, she is prone to falls.’*


The name of the specific medication was recorded in only 157 (13%) complaints. In many of the complaint issues, general references were made to medication classes instead; for example, the terms ‘antibiotic’ or ‘sedative’ were commonly used. This lack of detail in the complaint dataset may be because most consumers and representatives making complaints did not know the specific pharmacologic name of the agent. Table 3 shows the main medication agents mentioned in the dataset.

When the ‘APINCHS’ was applied for use in the dataset to broadly gauge the proportion of complaints issues relating to high-risk medications, over a third (n = 226 or 41%) related to high-risk medication use (Table 4). Of individual medicines specifically named in the complaint dataset, two thirds (n = 103 or 66%) included high-risk medication (Table 3). These high-risk medicines are shaded in Table 3 and Table 4.

### 3.4. Associations between the Complainant and Medication-Related Complaints

When medication-related complainants were compared to those who made all complaints in the 2019–2020 Annual Report, there was a significantly higher proportion of anonymous complainants, whereas considerably significantly fewer residents themselves made these types of complaints (Table 5). Over half of complaints were lodged by a resident’s representative or family; however, there was no difference between the proportion of representatives or family members, other interested parties, and external and internal agencies referrals making medication-related complaints and those making complaints overall.

## 4. Discussion

This research is the first known study to examine complaints made about medication in residential or long-term aged care in depth. Given the finding that medication-related issues received the highest number of complaints in RACS in the three Commission annual reports from 2018 to 2020, it is important to identify, categorize, and quantify them so deficits can be acknowledged and addressed [13,14,15].

It is important to appreciate that most of the medication-related complaint issues identified in this study relate to basic administration and management of medicines; they do not relate to more complex aspects of pharmaceutical care, such as drug interactions or adverse effects of medicines.

Not being given medication ‘at the right time’ was the leading complaint issue reported, accounting for over a quarter of the medication-related complaints received during 2019–2020 [19]. Complaint issues coded to this category describe consumers receiving medicine late, missing a dose, having medication withheld (often due to a lack of qualified staff), or not having a prescribed medication started at all. Notably, many of the complaints received refer to inadequate management of pain and palliative care, and poor control of Parkinson’s disease and diabetes—all conditions where the delivery of medication is time-critical for optimal effect and quality of life [7,19]. The fact that most complaints involved residents not receiving medicines on time or, in some cases, not having them started after they were ordered, points to a lack of staff and/or allocated time to administer medication as intended. It also reveals deficits in staff knowledge regarding the importance of giving medicines on time and not delaying treatment [7].

Likewise, the complaint category of ‘Poor overall processes, policies and procedures’, the second highest ranked medication-related complaint and the subject of over a fifth of all complaint issues, involved inadequate medication management [19]. Complaint issues raised in this category noted a lack of policies and systems, problems with storage and delivery, limited assistance to consumers who were taking medicines, and a dearth of qualified staff to administer ‘higher-risk’ medication, such as opiates and insulin. These findings replicate those in ‘The state of medication in New South Wales residential aged care’ report, in which nearly two thirds of aged care staff surveyed felt that their consumers experienced delays in acquiring pain relief and other essential medications [7].

These single two categories of medication-related complaints account for nearly half of all the complaint issues recorded. This finding strongly suggests there are not enough qualified staff in many aged care services to competently administer and manage medication. It also speaks to an apparent deficit in the quality of training on medication management and indicates a lack of accessible guidance for staff working in residential aged care. Urgent attention is needed to ensure those staff administering and managing medication have the appropriate qualifications and experience to ensure they can capably fulfil this important task. One of the recommendations in the recent Royal Commission into Aged Care Quality and Safety was to increase staffing of tertiary-qualified registered nurses (RNs) providing direct resident care [23]. Accordingly, legislation has now been amended to ensure increased RN staffing is provided from July 2023 [24]. Further research is needed to assess if enhanced staffing as proposed translates to improvements in overall medication management in this sector or is achievable given widespread nursing shortages.

Professional pharmacy bodies have long advocated for enhanced pharmacist involvement in RACSs [5,9,11], stating that the profession, as the experts on medication use, is ideally equipped to enhance medication management in this setting. Most focus to date has been on individual clinical medication review of the resident’s medicine regime as opposed to ensuring medicines are administered in a timely manner or service processes for medication administration and management [5,25]. Assistance with policy and procedure development for medication management processes, along with staff training, is currently available from pharmacists under the Commonwealth-funded ‘Quality Use of Medicine’ (QUM) program which is delivered to most Australian RACSs. Specifically, the QUM program aims to improve medication practices and procedures in RACSs [25] Activities that can be provided include (1) training the healthcare team on a range of issues, including medication administration; (2) participation in Medication Advisory Committees (MACs); (3) assistance with policy and procedure development; (4) assisting the service to meet medication standards and regulatory requirements; (5) assisting providers and residents to understand potential medication side effects, monitoring requirements, and review; and (6) supporting residents to self-administer medication when this is their preference and they are capable of doing so [25].

However, a 2018 external review of the QUM program was ‘unable to quantify the extent to which the program is leading to improvements and achieving its intended objectives’ [26]. The reviewers commented that the flexible nature of the current QUM program (in which pharmacists select the QUM activities delivered each quarter, in conjunction with providers), is likely to result in variations in outcomes, depending on what services are delivered [26]. Recommendations to strengthen the QUM program include ensuring that only activities with evidence for effect are funded, and the introduction of clear performance indicators to enhance accountability [26]. In 2022, the Australian Government announced new funding for on-site pharmacists to be based in residential aged care, with details of the program to be announced [27]. Furthermore, new mandatory medication management quality indicators on antipsychotic use and polypharmacy were introduced for RACSs in 2021 [28]. These initiatives with their emphasis on improving residential aged care quality and safety, including funding for embedded credentialed pharmacists, may assist to rectify the deficits in medication administration and management found in this evaluation.

The introduction of electronic medication management systems is likely to improve medication management in RACS and, thus, has the potential to reduce medication-related complaints. As the uptake of ‘e-systems’ in aged care increases, consumers with time-critical medication can be prioritized for medicine administration, and alerts provided to ensure doses are given optimally. In addition, processes and documentation associated with medication management would be enhanced.

The third most common complaint issue involves ‘chemical restraint’, in which medications are prescribed to limit a consumer’s movement and/or ‘manage’ their behavior [23,24]. The high use of psychotropic medication in Australian RACSs has been highlighted elsewhere [29]. The high frequency of reporting has most likely been influenced by the strong focus on restraint in the Royal Commission into Aged Care Quality and Safety [23]. One interesting observation is the number of complaints received about the lack of consent before sedating medications are taken accounting for over 5% of all medication-related issues. The frequency of these complaints suggests an increased awareness from family, or the consumer’s representative, about their legal right and concomitant responsibility in relation to shared decision-making, about the medical care of the person for whom they are legally responsible. In July 2019, the *Aged Care Act 1997* was amended to incorporate the ‘Quality of Care Principles, Part 4A’, which represented the first Australian legislation to regulate the use of restraint in aged care [24]. This legislation, along with quality indicators on antipsychotic use, offers considerable potential to moderate the use of chemical restraint in RACSs and thus, reduce the proportion of complaints in this category [24,28]. Future analysis of complaints in subsequent years may establish if the introduction of targeted legislation and closer scrutiny of antipsychotic use in residential aged care improves informed consent processes, monitoring, and oversight of chemical restraint practice, with a resultant reduction in the proportion of complaints about their use. It is also important to note that ongoing increased awareness of residents and their families about chemical restraint and the legal requirement for informed consent may also lead to a rise in these types of complaint issues.

### 4.1. Resident and Family Involvement in Medication Management

When the cohort or participant group of those lodging complaints (complainants) about medicines was compared to all those who made complaints in the 2019–20 Annual Report, significantly fewer residents themselves, less than 9% of complainants, lodged these complaint issues. This finding may indicate, as other researchers have observed, that residents in residential aged care have limited involvement in their medication management [30,31]. Residents taking medication may also have limited ability to engage in the complaints process themselves due to cognitive or sensory impairment. They also may not be aware of their right to complain.

Many of the complainants, particularly relatives, reported that their consent was not sought by prescribers when new medication, particularly sedating medication, was prescribed. Likewise, many were not involved, or even told, when changes were made to existing medicine regimens. A recent systematic review on resident and family engagement in RACSs concluded that planning around medication management was opportunistic rather than intentional, and that residents and families were rarely actively involved in decision-making processes [30]. Communication between providers, prescribers, residents, and their families tended to be unidirectional, involving mostly medication information provision, and the preferences of residents and families were rarely sought or enacted [30,31].

The ‘Guiding principles on medication management in Residential Aged Care Facilities’ was originally published in 2012 and has been recently updated [2,32]. A feature of the revised guidelines is a strong emphasis on involving residents and relatives in decision-making around medication management, with the first guiding principle now endorsing ‘person-centered’ care and the second, ‘communication about medicines’. The guiding principles endorse that ‘each resident, including their carer, family and/or substitute decision-maker is supported in a safe, respectful, and appropriate manner to navigate the assessment and consent processes involved in the use of medicines’ [2]. The release of the updated principles is both timely and relevant, as the number of complaints about the lack of shared decision-making in this evaluation attests.

### 4.2. Limitations and Strengths of this Research

One limitation of the medication-related dataset is that as all the keywords and sub-keywords in NCCIMs were not examined in depth, all complaints about medication may not have been captured. However, a strength of the research is that the coding framework developed for this project was comprehensive enough to categorize all the complaint issues described. Ideally, this framework should be re-used in subsequent years with this report adopted as a benchmark, to gauge the effectiveness of strategies implemented to address deficits in medication administration and management, and to detect emerging trends.

Another important limitation of the dataset is that many of the complaints lacked key clinical details about the medication used, dose, duration of use, and indication. This lack of detail is likely due to low levels of health and medication literacy of residents and their families [30]. This observation suggests that these cohorts were not given basic information about medicines prescribed. Furthermore, the frequency of complaints about the lack of informed consent sought attests to residents and their representatives not being involved or informed when therapeutic decisions were being made [30]. More research is needed to determine why residents and families are often effectively bypassed and not involved in the decision-making process around medicines, and why their opinions are not sought about this important aspect of clinical care.

A final limitation of this dataset is that reporting of complaints may have been affected by the COVID-19 pandemic which started in late March 2020 and resulted in RACS lockdowns, staff shortages, and visitor and service restrictions. The frequency, type of complaint issues, and complainant is likely to have been impacted in the last three months of reporting in 2020, within the timeframe of this evaluation.

The key strength of this research is that the voices and opinions of the consumer, relatives, nursing and care staff, and others who are not routinely heard are captured, listened to, and examined, and their concerns about medication management and administration are identified, highlighted, and addressed.

## 5. Conclusions

The main complaint issues lodged in relation to medication management and administration in Australian residential aged care during 2019–2020 related to not receiving medicines on time, inadequate medication management systems, policies and procedures, and chemical restraint. The medical conditions or indications about which the highest number of complaints were received involved pain management and/or palliative care. Family members or representatives were the main participant group or complainant type making medication-related complaints and this was observed with complaints overall. However, significantly fewer residents themselves made medication-related complaints and there was a greater proportion of anonymous complainants. This complaint database analysis has provided rich data about medication-related complaints from a large and representative sample to provide data that are highly consumer-focused and relevant. The ongoing analysis of medication-related complaints will provide useful insights not only into the performance of aged care providers, but will also assist in informing and potentially enhancing the medication health literacy of care recipients and their family members or representatives.

## Figures and Tables

**Figure 1 pharmacy-11-00063-f001:**
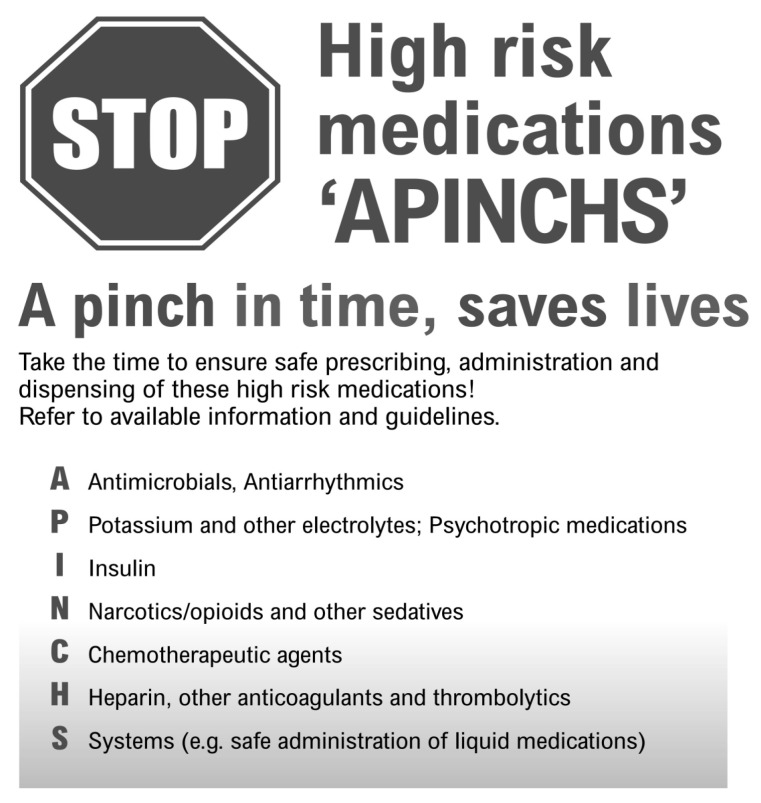
The APINCHS categorization system [21].

**Figure 2 pharmacy-11-00063-f002:**
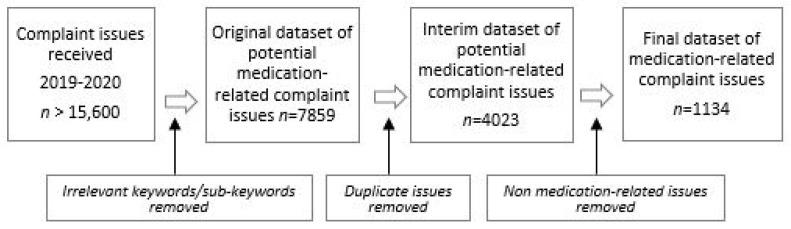
Process chart of dataset development [15].

**Table 1 pharmacy-11-00063-t001:** NCCIMS keywords and sub-keywords included in medication-related dataset extract.

Keyword	Sub-Keyword	Issues
Abuse	Physical	8
	Psychological/emotional	1
Choice and dignity Client assessment & implementation Consultation and communication	Right to refuse medication	7
	Polypharmacy review	3
	Ability to express needs/wants	3
	Information about medication	31
Goods & equipment	Medical and pharmaceutical supplies	12
Healthcare	Allied health assessment and services	10
	Chemical restraint	77
	Constipation and continence management	22
	Dementia management	14
	Falls prevention and post-falls management	9
	Infectious diseases and infection control	23
	Medication administration/management	824
	Pain management	60
	Palliative/end-of-life care	18
	Behaviour management	11
Personal care	Personal safety and interventions	1
Total		1134

**Table 2 pharmacy-11-00063-t002:** Coding framework of the types and frequencies of medication-related complaints *.

Keyword	Sub-Keyword	Categorization	Complaint Issue	No. Issues
Healthcare	Medication administration and management Nursing Chemical restraint	Right resident Right medicine Right dose Right time Right route Right documentation Monitoring Appropriate policies/ systems Appropriate policies/ systems Restrictive practice	- giving resident another’s medicine - staff giving incorrect medicine - given medicine they are allergic to - wrong dose given - medicine missed or withheld - medicine given late - medicine delayed or not started - medicine given incorrectly - medication given unlawfully - medication charts not accurate - adverse effects missed - residents not monitored for effect - inadequate policies and systems - poor storage and accountability - poor ‘prn’ medication policy - tablets found on floor or untaken - medicines given by unqualified staff - self-administration not facilitated - medication used as restraint	25 36 10 **53** **119** **113** **80** 27 44 38 30 22 **116** 30 16 13 **68** 28 **168**
Choice & Dignity	Right to refuse medicine	Right to refuse medication	- informed consent not sought - forced to take medication	65 13
Consultation & Communication	Medication Information	Medication information	- no information at care transitions - not informed when changes made	24 19
Service Implementation	Polypharmacy review	Access to medication review	- medication regimen not reviewed	13
Goods & equipment	Pharmaceutical supplies	Quality provision of medicine	- substandard provision of medicine - poor pharmacy supply service	47 7
Total				1224

* Those issues receiving over 50 complaint issues have been emboldened for emphasis.

**Table 3 pharmacy-11-00063-t003:** Medical conditions associated with medication-related complaint issues ^1^.

Medical Condition	Additional Details	No. of Issues	Total no. of Complaint Issues
Pain	S8/opioids	91	181
management/	Paracetamol	8	
palliative care	S4 medication	4	
	General pain management	78	
Sedation	Antipsychotics	35	172
	Benzodiazepines	16	
	Agent not specified	121	
Infectious	Urinary tract infection (UTI)	28	54
disease	Respiratory infections	4	
	Fungal infections	4	
	Skin infections	3	
	Antibiotics in general	15	
Diabetes	Insulin	16	29
	Other diabetes issues	13	
Parkinson’s disease	Levodopa	6	28
	Apomorphine	2	
	Late or missed doses	20	
Asthma	Inhalers and nebulizers	18	18
Constipation	Laxatives	16	16
Eye	General eyedrop use	14	14
Anticoagulation	Warfarin or ‘blood-thinner’	8	11
	Aspirin	1	
	INR results	2	
Vaccination	Influenza	10	10
Cardiovascular	Heart	5	10
	Hypertension	3	
Immunotherapy	Oedema	2	
Anticonvulsants	Corticosteroids/chemo	6/2	8
Thyroid	Valproate/carbamazepine	3	2
	Thyroxine	1	1
Total			554

^1^ High risk medicines shaded.

**Table 4 pharmacy-11-00063-t004:** Main medicines associated with medication-related complaint issues ^1,2^.

Medicine	No of Complaint Issues
Morphine	31
Oxycontin	22
Risperidone	16
Insulin	16
Influenza vaccine	10
Paracetamol	8
Levodopa	6
Buprenorphine	4
Fentanyl	4
Warfarin	4
Quetiapine	3
Oxazepam	3
Prednisolone	3
Salbutamol	3

^1^ Only those medicines mentioned in three or more complaint issues are included. ^2^ High risk medicines shaded.

**Table 5 pharmacy-11-00063-t005:** Complainant group comparison.

Complainant Type	No Medication-Related Complaints	Proportion of Complaints (%)	No of Complaints 2019–2020 Report [15]	Proportion of Complaints (%)	Proportion Difference (%)	Chi- Squared	*p* Value *
Representative or family	622	54.9	4507	52.8	2.1	1.8	0.183
Anonymous	306	27.0	1569	18.4	8.6	47.3	<0.0001
Consumer	96	8.5	1686	19.7	−11.2	83.7	<0.0001
Other interested	89	7.9	602	7.0	0.9	1.1	0.299
External agency/ Internal referral	20	1.8	175	2.1	−0.4	0.8	0.372
Total	1134	100	8539	100			

* Significance level was set at *p* < 0.05 (significant findings—bold text).

## Data Availability

The medication-related complaints dataset is archived and secured at the Commission. For further information please contact kathleen.williams@agedcarequality.gov.au.

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
