# Peer review of "Medication-Related Complaints in Residential Aged Care"

_pharmacy, 2023, doi:10.3390/pharmacy11020063_

Round 1
Reviewer 1 Report
Table 1: the horizontal lines would easy for understanding the information.
Figures 1 and 2 – can be included in a supplementary file, not the main manuscript
Row 187 and 318 – abbreviations RN and NSW should be explained
Table 3 - the numbers in the “No issues” column should be in line with the corresponding Complaint Issue
Table 3 – the reason of using bold should be explained
Row 203 – n=312 is not 26% of 1224
Row 223: a reference should be included regarding the drugs enrolled in Schedule 8 category; in Europe another classification system (https://www.emcdda.europa.eu/publications/topic-overviews/classification-of-controlled-drugs/html_en) is used.
Regarding the complaints, because it is not possible to check the verity of the statements; the approach is somehow a one-sided (E.g., row 242-244: why are we sure that the family was not consulted?).
Row 304: The sentence “They are not complex.” It is not obvious what does it refer to?
Row 385: The sentence is not finished
Reviewer 2 Report
This is a very nicely written paper. I found myself making only minor comments that are only stylistic. I recommend acceptance.
Author Response
Dear reviewer,
Thank you for your kind review and acceptance. The other reviewers have recommended some alterations to the method, formatting and discussion. I hope the amendments are acceptable to you.
Regards,
the authors
Reviewer 3 Report
Introduction: Presented well the need for the study.
Methods: There are many concerns about the flaws in this section.
For example, the authors do not explain how the data was collected for the “complaints.” Who filled out the forms, storage of data, and what type of questions were asked.
What type of analysis was conducted and why? The authors stated that the usage of the Chi-Square.
Was the data collection anonymous?
The authors should have stated on how long and in-depth the answers were for the qualitative part.
The discussion needs to discuss the results of the study. For example, the first sentence does not summarize this study's results and novelty. Furthermore, the following sentence needed to be more concise and explain the purpose of this study.
Line 305 "at the right time," does not provide references, and the sentences are lengthy and lack cohesion. The entire paragraph provides details without any references. Furthermore, the authors do not offer an argument for their discussion.
Lines 337-342 do not follow the format of a discussion. Please remove the bullet points and write the discussion as a paragraph.
Limitations: Although the authors mentioned the limitations of this study, the authors did not address any strengths of this study.
Line 414-419: This paragraph needs to be clarified—more information regarding statistics and how this study brings a novelty. Furthermore, the study does not provide any need for future research. Thus, a few sentences that connect the limitations with future research must be addressed.
Round 2
Reviewer 3 Report
Methods
· The authors failed to mention what questions are on the complaint form. The manuscript would greatly benefit to either add the questions in detail to the article or to attach a copy of a blank form in the appendix.
· It can be seen that the exclusion criteria is separated between lines 115-116 and 130-133. It would be beneficial that they be brought together or put in a table.
· The removal of lines 149-157 would improve the article as they are not relevant to the study.
· It was stated in lines 167-168 that the results were tallied and recorded, however the authors failed to state how this was done.
· An additional point of weakness would be that the statistical analysis section is analyzing complainant type, which was not mentioned in the introductory endpoints and therefore has little relevancy.
Results
· Overall, the authors did well on summarizing the results from the analysis and provide good examples for the topmost common complaint issues mentioned.
· Line 304, word choice: “detail” – can be changed to “include,” “contain,” or “mention.”
· Line 305, not clear on what the authors try to say by “The conditions about which the highest number of complaints were received were ‘pain management and/or palliative care’.”
· Table 6: second and fourth columns: “No” and “No of” can be dismissed or changed to “No. of”
Discussion
· Good summary on the findings as well as providing current regulations related to each of these gaps. However, authors need to provide some extrapolation on how the issues mentioned in the complaints can create gaps in healthcare in residential aged care services (RACS) which in turn create errors or problems in the RACS, and some solution options to these problems.
· Line 417 and 440: word choice: “may go some way” – need to change the terms used and clarify meaning to specify whether further study would bring benefits or what.
· Line 431 and 432: “One interesting observation is the number of… these medications are taken.” – providing the actual number or percentage in here could provide readers a mor complete image of how common this finding is.
· Line 442 and 471: “4.1” and “4.3” but there is no “4.2”
· Line 443: “complainant type of those lodging” – what does this mean?
· Line 474 to 477: this part seems to convey the strength of the “Coding Frame” rather than the study itself.
Conclusion
· The conclusion provides an overall characteristic of complaint issues, but did not indicate the findings from this specific study/analysis.
